# Conservation Agriculture in Semi-Arid Zimbabwe: A Promising Practice to Improve Finger Millet (*Eleusine coracana* Gaertn.) Productivity and Soil Water Availability in the Short Term

Vengai Mbanyele [1,2,*], Florence Mtambanengwe [1,2], Hatirarami Nezomba [1,2], Jairos Rurinda [1,2] and Paul Mapfumo [1,2]

1 Department of Soil Science and Environment, University of Zimbabwe, Mount Pleasant, Harare P.O. Box MP167, Zimbabwe; fmtambanengwe@admin.uz.ac.zw (F.M.); hnezomba@agric.uz.ac.zw (H.N.); jairurinda@agric.uz.ac.zw (J.R.); pmapfumo@admin.uz.ac.zw (P.M.)
2 Soil Fertility Consortium for Southern Africa (SOFECSA) Research Group, University of Zimbabwe, Mount Pleasant, Harare P.O. Box MP167, Zimbabwe
* Correspondence: vengai.mbanyele@students.uz.ac.zw

**Abstract:** Increasing within-season dry spells in Southern Africa in recent years have generated growing interest in conservation agriculture (CA) to secure crop yields, especially under rainfed systems. This study aimed to evaluate the effects of CA on finger millet's (*Eleusine coracana* (L.) Gaertn) growth, yield and water use efficiency on nutrient-depleted sandy soils. Five treatments, namely (conventional tillage (control), conventional tillage + mulch (partial CA1), reduced tillage only (partial CA2), reduced tillage + mulching (partial CA3) and reduced tillage + mulching + intercropping (full CA)) were evaluated over two consecutive cropping seasons (2015/16 and 2016/17) on-farm in the village of Chidora in Hwedza District, southeast Zimbabwe. All mulched treatments had 15–32% more soil water content over the two growing seasons compared to the control. The higher soil water content under the mulched treatments significantly improved finger millet growth and development during both seasons as evidenced by the lower number of days to emergence (3 days less), greater shoot biomass, higher number of productive tillers and higher number of fingers produced. The full CA treatment achieved the best finger millet grain yield of 1.07 and 1.29 t ha$^{-1}$ during the 2015/16 and 2016/17 seasons, respectively. Full CA, partial CA3 and partial CA1 increased finger millet grain yield by 70%, 14% and 17% during the 2015/16 cropping season compared to the control. During the 2016/17 cropping season, a similar trend in finger millet grain yield was observed. Full CA was also among the most efficient methods in terms of water utilization (WUE), especially during the 2015/16 season. We concluded that CA, particularly when practiced in full, was more effective at offsetting the water limitations imposed by intra-seasonal dry spells on finger millet and significantly improved productivity.

**Keywords:** dryland cropping; mulching; finger millet; dry spells; yield; water use efficiency

## 1. Introduction

Rainfed agriculture is the world's biggest biome and crucial for food production especially in sub-Saharan agriculture [1]. However, water deficit conditions imposed by intra-seasonal dry spells pose a major crop production constraint, particularly when combined with poor nutrient management [2,3]. Increased incidences of these dry spells in Southern Africa due to climate change [4] means increased vulnerability of rainfed crops. While irrigation has great potential to supplement rainwater during dry spells to avoid significant crop yield loss or complete crop failure, the majority of farmers, especially smallholders, have no access to irrigation. In order to reduce the adverse effects of these dry spells, agronomic strategies that are within farmers' reach, e.g., tillage, timing of planting and selection of crop types and varieties, among others, have been in the limelight.

Several studies suggest traditional cereals, such as finger millet (*Eleusine coracana* Gaertn.), sorghum and pearl millet, which are more tolerant, particularly to rainfall and temperature changes, are more critical than commonly grown maize for improved food security [5–8]. In countries such as India, the largest producer of finger millet in the world [9], finger millet production and consumption has traditionally been part of its cropping systems. Likewise, in sub-Sahara Africa (SSA) and other parts of Asia, traditional grains, particularly millets, have been integral in supporting household food needs in times of climatic shocks and similar circumstances [6]. While government policies in most countries generally favouring production and consumption of fine cereals, such as maize, rice and wheat, compared to traditional millets, the increasingly changing climate means it is highly unlikely that such climate-sensitive crops will be able to support future food needs. However, just like maize, drought conditions are by no means the optimum conditions for achieving the best yield in finger millet. With the human population rapidly increasing in the region, continuing to improve rainfed crop production in drought-prone semi-arid areas through optimizing soil water management is thus critical for guaranteed food security and sustainability.

The growing of finger millet in drought-prone environments in Africa is not only important for food and nutritional security but also production of feed, starch and beverages. It is against this background that many countries in SSA, and other developing nations, are championing production of traditional grains, particularly in the wake of changing climate, for building resilient cropping, food and feed systems. Supportive policies at the regional level, such as the "Commodities strategy flagships" under the African Union Agenda 2063 seek to promote production, processing and marketing of traditional grain crops based on the value chain approach [10]. At the national level, the Government of Zimbabwe is promoting production of traditional cereal grain crops through national programs such as the CA-anchored "Pfumvudza agriculture" [11]. Finger millet is a rich source of valuable amino acids, especially methionine, which is often lacking in the diets of many poor people who live on starchy staples such as maize and cassava [12]. High amounts of iron, zinc, calcium, potassium, magnesium and vitamins in finger millet grain have also been reported [13]. Moreover, the low glycaemic index and slow digestion of finger millet foods due to their high fibre content makes them ideal for diabetic patients [12]. Most traditional grains grown in SSA are often low-yielding (often producing less than half of their potential yield) mainly due to drought, poor nutrient management and the use of poor unimproved landraces [14,15]. Despite finger millet being regarded as less demanding in terms of nutrients [16], improved soil fertility management, including growing nitrogen-fixing grain legumes in rotations or intercrops with the cereal, has been shown to significantly improve yield, as shown by, e.g., Rurinda et al. [15]. It is also important to quantify the grain yield response of finger millet to improved water management in view of the crop being generally characterized as less water-demanding.

Conservation agriculture (CA) has been promoted in Southern African countries, including Zimbabwe, as one of sustainable climate smart technologies capable of counterbalancing the water deficit conditions imposed by dry spells [17,18]. Three principles, namely reduced tillage, mulching through covering the soil surface with at least 30% crop residues and diversified cropping systems that include a legume crop either in rotations or mixtures, form the basis of CA [19]. Conservation agriculture may greatly improve rainfed crop yields due to increased soil moisture, reduced water loss and sometimes topsoil temperature modification from the mulch cover [20–22]. Not all smallholder farmers in SSA practice all the three CA principles. Some farmers choose to practice conventional tillage, instead of reduced tillage, which is often associated with the emergence of permanent weeds, according to Giller et al. [23], while embracing mulching and crop diversification principles. Some may choose to drop mulching because available crop residues are used for feeding livestock, but in practice reduce tillage and crop diversification [20]. Crop diversification can be a challenge for some farmers; hence, they may choose to only concentrate on the cereals that are guaranteed to be marketable as staple food [24]. It is therefore key to adapt CA to local circumstances. The objectives of this study were as follows: (1) to

evaluate the short-term impact of CA in both partial and full practice of principles on soil moisture dynamics under finger millet; (2) to determine the changes in crop growth and developmental processes that contribute to yield improvement; and (3) to determine water resource use efficiency of various treatments. This research could be helpful for soil water management strategies in rainfed finger millet production in semi-arid areas.

## 2. Materials and Methods

### 2.1. Study Area

This study was conducted over two successive cropping seasons (2015/16 and 2016/17) at a representative site in Hwedza District, southeast Zimbabwe. Traditionally, the district is largely a smallholder farming area dating back to over 100 years ago. Each farmer owns 2 to 5 ha. Annual rainfall during a normal season in Hwedza, which starts in early November and ends in early April, ranges between 650 and 800 mm. Maize is the major staple crop and often allocated the largest area under production. In the past two decades, the area has increasingly experienced poor seasonal rainfall distribution at both temporal and spatial scales, with prolonged intra-seasonal dry spells causing persistent maize failure. The effect of dry spells on crops is exacerbated by the poor water holding capacity of the dominantly sandy soils derived from the granite rock. With persistent maize failures in recent years, a number of farmers in the area are diversifying into traditional crops, such as finger millet, that are perceived to be more drought-tolerant in order to stabilize household food supplies. However, the yields achieved are generally way below the potential yield, and often vary significantly from season to season. Based on crop assessment surveys conducted by the Department of Agricultural Research and Extension of the Zimbabwe Ministry of Lands, Agriculture, Fisheries, Climate and Rural Development (AGRITEX), they rarely exceed $0.5$ t ha$^{-1}$ on average, as compared to the attainable potential of approximately $2.5$ t ha$^{-1}$. The farming system in the area is predominantly mixed crop–livestock, with cattle being a key component in crop production through provision of manure and draught power. Farmers in the area typically practice conventional tillage through ploughing the whole field using the ox-drawn mouldboard plough. Some practice conservation agriculture (CA) but mainly under maize production. Common CA principles that are applied singularly or in in combination are reduced tillage through ripping and planting basins, mulching using crop residue or grass and intercropping with cowpea.

### 2.2. Selection of Treatments and Experimental Site

A focus group discussion (FGD) was held to select treatments and a suitable site for testing the selected treatments. The FGD consisted of 10 men and 10 women. All the participants had farming experience in the area that exceeded 30 years. The community leadership (i.e., village heads, headmen) and local agricultural extension personnel were also included. They were first divided into women only and men only groups and later combined to build consensus. Conventional tillage (common farmer practice), conventional tillage with mulching, reduced tillage only, reduced tillage with mulching and reduced tillage, mulching and intercropping combination were the five treatments agreed upon for testing. It was also agreed that the experiment would be established on sandy fields typical of the local condition where finger millet is grown. As a result, transect walks, guided by the participants, were carried out to identify and select a field that was accessible and large enough to accommodate all the treatments. The experiment was therefore established at Chidora Farm in Makwarimba Ward in Hwedza District, southeast Zimbabwe (18°39′ S 31°37′ E, 1409 m above sea level (m.a.s.l.)) (Figure 1). The farm is owned by the village head and is a strategic learning centre for over 30 households in the village. The soil at the experimental site is granite-derived sand that is inherently low in organic carbon and has poor water holding capacity. The soil is typical of smallholder farms in Zimbabwe. Initial characterization of the soil at the experimental site was performed by taking 10 samples up to a 0.45 m depth along an X shape covering the experimental field area. The sub-samples were thoroughly mixed to obtain a composite sample. The composite sample was air-

dried and sieved using a 2 mm-mesh sieve and analyzed for texture (hydrometer method), pH (0.01 M CaCl$_2$), organic carbon (modified Walkley–Black method), available P (Olsen method) and total N (micro-Kjeldahl method) [25]. A core was used to take undisturbed soil samples for bulk density determination. The soil analytical results are given in Table 1.

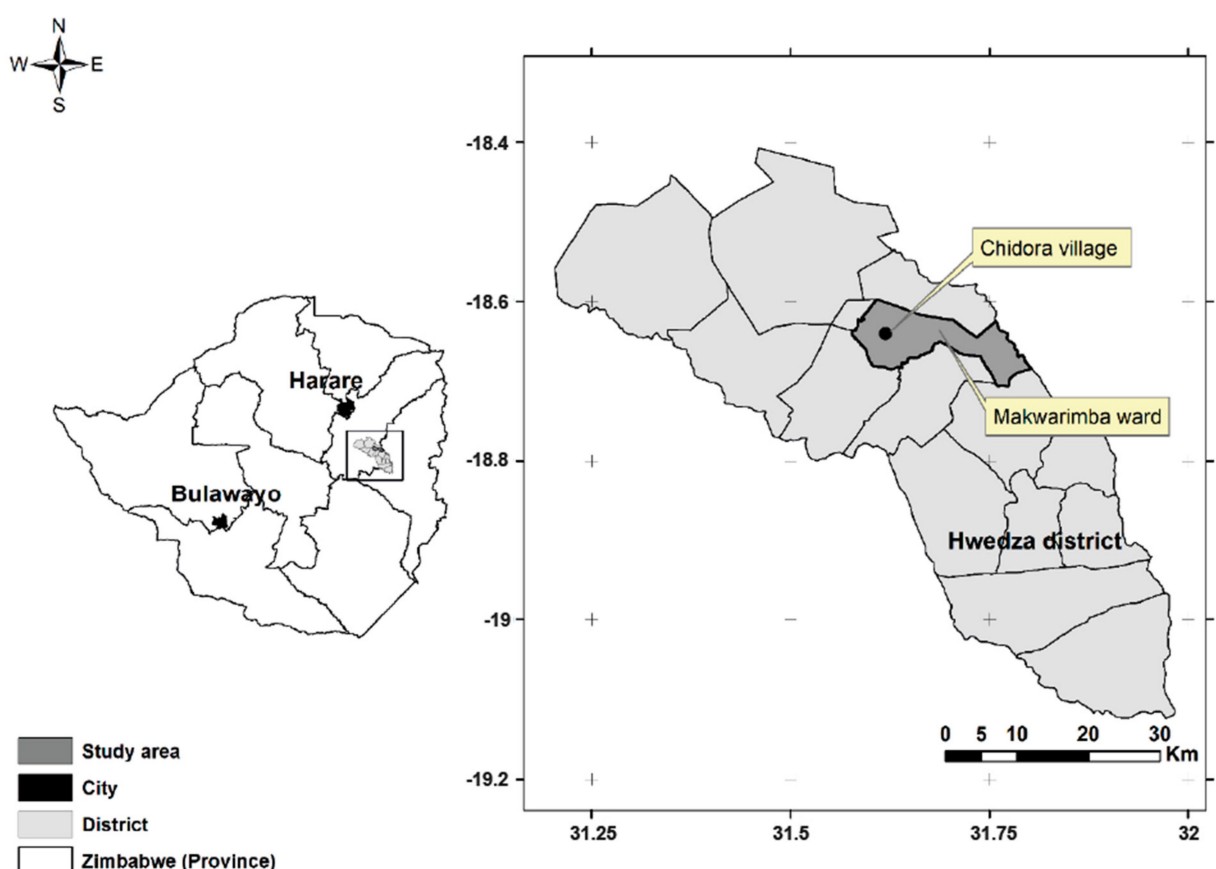

**Figure 1.** Experimental site location.

**Table 1.** Initial topsoil (0–45 cm) characteristics at experimental fields.

| Soil Parameter | Value |
| --- | --- |
| Physical | |
| Bulk density | 1.58 |
| Sand (%) | 75 |
| Clay (%) | 9 |
| Chemical | |
| pH (0.01 M CaCl$_2$) | 4.66 |
| Total N (%) | 0.03 |
| Available P (mg kg$^{-1}$) | 3.80 |
| K (cmol$_{(c)}$ kg$^{-1}$) | 0.18 |
| SOC (%) | 0.47 |

*2.3. Experimental Design and Management*

The following five treatments were selected for testing: (i) finger millet under conventional tillage only (common farmer practice), (ii) finger millet under conventional tillage + mulching (partial CA1), (iii) finger millet under reduced tillage only (partial CA2), (iv) finger millet under reduced tillage + mulch combination (partial CA3) and reduced tillage + mulch + finger millet–cowpea intercrop (full CA). The treatments were evaluated during both seasons on the same plots. An animal-drawn mouldboard plough

was used to till the whole plot to a depth of approximately 25 cm just before planting under conventional tillage. For reduced tillage, a ripper tine attached to the beam of an ordinary ox-drawn mouldboard plough was used to open rip lines approximately 25 cm deep. Sun-dried locally available thatching grass (*Hyparrhenia filipendula* (L.) Stapf was applied at 2.5 Mg ha$^{-1}$ (on a dry weight basis) soon after planting in all mulched plots. This was enough to achieve approximately 30% soil surface cover as per recommended CA practice. Grass is readily available in the area and was chosen ahead of commonly used maize residues in several CA studies. A local finger millet landrace called ''Gweza'' and a semi-erect cowpea cultivar (CBC2) (for the intercropping treatment) were used as the test crops. All the treatments involved planting finger millet at 0.45 m × 0.10 m, resulting in population densities of 220,000 plants ha$^{-1}$. For the intercrop treatment, cowpea was sown between the finger millet rows two weeks later to reduce adverse effects of competition on the primary crop (finger millet), especially in the early crop growth stages. This was also strategically done to spread out labour. Treatments were maintained in the same plots during the second season with the grass used for mulching obtained from the same source (i.e., field edges). Typical fertilizer rates were used in all plots with basal mineral fertilizer, compound D (7% N, 14% P$_2$O5 and 7% K$_2$O) applied at planting at 230 kg ha$^{-1}$ and ammonium nitrate (34.5%) as top dressing applied in two splits of 45 kg N ha$^{-1}$ each. The treatments were applied to 64 m$^2$ (8 m × 8 m) plots arranged in a completely randomized block design with three replicates. While initial weed control under conventionally tilled plots was carried out through ploughing the whole plot, in reduced tilled plots, glyphosate (N-(phosphonomethyl) glycine) herbicide was sprayed at 3.5 L ha$^{-1}$ before sowing. The hand hoe was then used for subsequent weed control in all plots. On reduced tilled plots, hand hoeing was carefully performed through scratching on the soil surface to avoid disturbing the soil during weeding.

*2.4. Sampling and Measurements*

2.4.1. Determination of Soil Water Content and Evapotranspiration (ET)

Soil cores were used to collect soil samples at three random locations throughout the area of each plot for determination of soil volumetric water content (%) during the two seasons. Soil samples were collected every seven days after every rainfall event. They were collected to a depth of 0.45 m in three incremental depths of 0–0.15, 0.15–0.30 and 0.30–0.45 m. Immediately after collecting, the samples were packed in air- and water-tight zip-lock plastic bags for transportation to the laboratory. In the lab, the soils were weighed and oven-dried at 105 °C for three days until a constant dry weight was reached. Volumetric moisture content was calculated as a product of gravimetric water content, bulk density and soil layer thickness.

Seasonal evapotranspiration (ET, mm) for individual plots was determined according to Equation (1).

$$\text{ET} = \text{R} + \Delta\text{SWC} \tag{1}$$

where R is the total rainfall received (mm) and $\Delta$SWC is the change in soil water content (mm) between the two sampling times. There was no irrigation water applied. Rainfall was therefore the sole source of water and was measured at the experimental site using an ordinary standard rain gauge, which was mounted 1 m above the ground. Ground water recharge, deep drainage and runoff were assumed to be negligible. Such assumptions have been made under similar conditions by Miriti et al. [26].

2.4.2. Plant Measurements and Calculation of WUE

The number of days taken for at least 50% of plants to emerge (based on expected plant population) was recorded for every plot. The vegetative stage was from emergence to flowering. The flowering stage was when 50% of the plants in the plot showed panicles. During the growing season, six adjacent finger millet plants were sampled fortnightly to determine shoot biomass and the growth rate (i.e., weight gain over time). The selected plants were cut just above the ground and dried in a fan oven at 70 °C to a constant weight

before weighing. The total shoot biomass in each plot was expressed as kg ha$^{-1}$. Six adjacent plants were also selected within each plot and marked to count the number of tillers as well as the number of fingers and finger length at maturity. Plants that had a "fist-like" panicle were counted in the whole plot, and that number was expressed as a percentage of the plant population. In Hwedza and smallholder farming areas in Zimbabwe, finger millet that produces a "fist-like" panicle is the most desirable as it shows that the crop grain developed well without being stressed by yield-limiting factors such as water, nutrients, sunlight and pests and diseases. The "fist-like" panicles are often retained as the seed for the next cropping season. At maturity, grain yield was measured for all plants selected from a 25 m$^2$ area in each plot. The grain yield was determined at 12.5% moisture content based on the mean of the three replicates. To obtain water use efficiency (WUE), Equation (2) was used:

$$\text{WUE (kg/mm)} = \frac{\text{Grain yield (kg / ha)}}{\text{ET(mm)}} \tag{2}$$

where WUE = water use efficiency (kg mm$^{-1}$ ha$^{-1}$); and ET = evapotranspiration (mm).

Only finger millet grain was considered for WUE in intercrop.

### 2.5. Statistical Analyses

The effects of the treatments on the measured parameters for finger millet crop were evaluated using a one-way analysis of variance (ANOVA) in Genstat. This analysis focused on finger millet performance and excluded cowpea, which was only used in one treatment as a component crop in intercropping. Data were analyzed by individual season to assess the performance of the tested treatments. However, season was considered a fixed factor for grain yield only with the split-plot design used with season as the main plot and treatments as the sub-plot. Correlation analysis was used to assess relationships among yield and total tillers, productive tillers, average number of fingers per panicle, finger length and "fist"-like fingers. In all cases, differences were deemed to be significant at $p < 0.05$, and the least significant difference (LSD) was used to separate the means.

## 3. Results

### 3.1. Soil Water Dynamics

Volumetric soil water content ranged between 5 and 32% across treatments during the 2015/16 cropping season (Figure 2a). The season had a lower cumulative in-crop rainfall amount (350 mm) and experienced higher incidences of intra-seasonal dry spells (Figure 2c). Partial CA1, in which conventional tillage was combined with mulching, achieved the greatest volumetric soil water content ($p \leq 0.05$) up to 60 days after planting (DAP), while partial CA2 (reduced tillage only) had the least throughout the season (Figure 2a). The greatest volumetric water content of 32%, for partial CA1, was achieved early in the season 9 DAP (Figure 2a). Mulching increased volumetric soil water content by 33–55%, particularly during the early part of the season up to 60 DAP. From 60 DAP up to harvesting, when mulching cover was no longer visible in most mulched plots, the full CA treatment (reduced tillage + mulching + intercropping) achieved the greatest volumetric water content, while partial CA2 continued to exhibit the worst performance. During the same period (i.e., 60 DAP), the differences among the control, partial CA1 and partial CA3 were not significant. During the 2016/17 season, the volumetric soil water content ranged from 8 to 36% (Figure 2b). The 2016/17 cropping season had more cumulative in-crop rainfall (1115 mm) and experienced lower incidences of intra-seasonal dry spells (Figure 2c). Full CA had the best soil water content throughout the season, although there was no significant difference resulting from the partial CA1 treatment in most instances (Figure 2b). Similar to the 2015/16 season, partial CA2 had the worst performance. The highest volumetric soil water content of 36% was achieved at 110 DAP. Mulching almost doubled the volumetric water content in both tillage practices, especially up to 100 DAP.

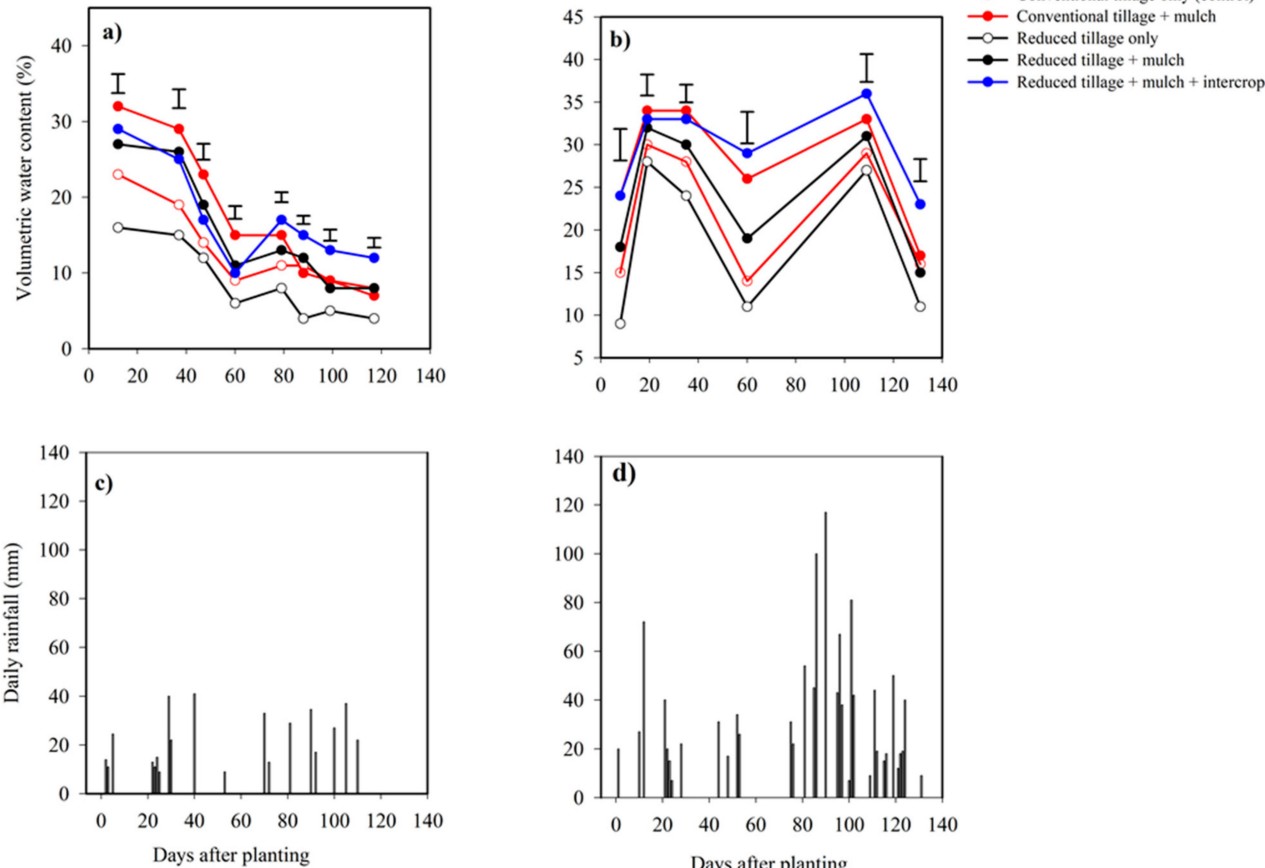

**Figure 2.** Volumetric water content (0–45 cm) (**a**,**b**) and daily in-crop rainfall (**c**,**d**) under conventional tillage (control), conventional tillage + mulch, reduced tillage only, reduced tillage + mulch and reduced tillage + mulch + intercropping with cowpea treatments in 2015/16 and 2016/17. Error bars represent LSD$_{0.05}$.

*3.2. Finger Millet Development*

Mulching significantly promoted early seedling emergence by 3 days during both seasons (Figure 3). In all mulched plots, 50% of seedling emergence was attained at 10 DAP while it was at 13 DAP for non-mulched treatments. During 2015/16 season, the vegetative stage was the longest under full CA (72 days). In the order from the longest to shortest vegetative stages, the treatments demonstrated the following stage duration: partial CA1 (69 days) > partial CA3 (65 days) > control (63 days) > partial CA2 (62 days). The same trend was maintained for the reproductive period with full CA achieving the longest reproductive period (59 days). This was 37% and 26% more than the control and partial CA1, respectively. While there were no significant differences in the influence of any of the treatments on vegetative stage length during the 2016/17 cropping season, the reproductive period was extended by 4 days under full CA practice.

With regard to the number of tillers, number of fingers, finger length and number of "fist-like" panicles, full CA performed the best, followed by partial CA1 in most cases (Table 2). Considering the number of "fist like" panicles, a local indicator of well-developed grain due to optimum environmental conditions for crop growth, full CA presented the best conditions, as depicted by over 75% of the plants showing such a characteristic during both seasons (Table 2). This translates to over 60% more than the control. All the parameters in Table 2 showed a strong correlation with yield with $R^2$ values ranging between 0.65 and 0.91 (Table 3).

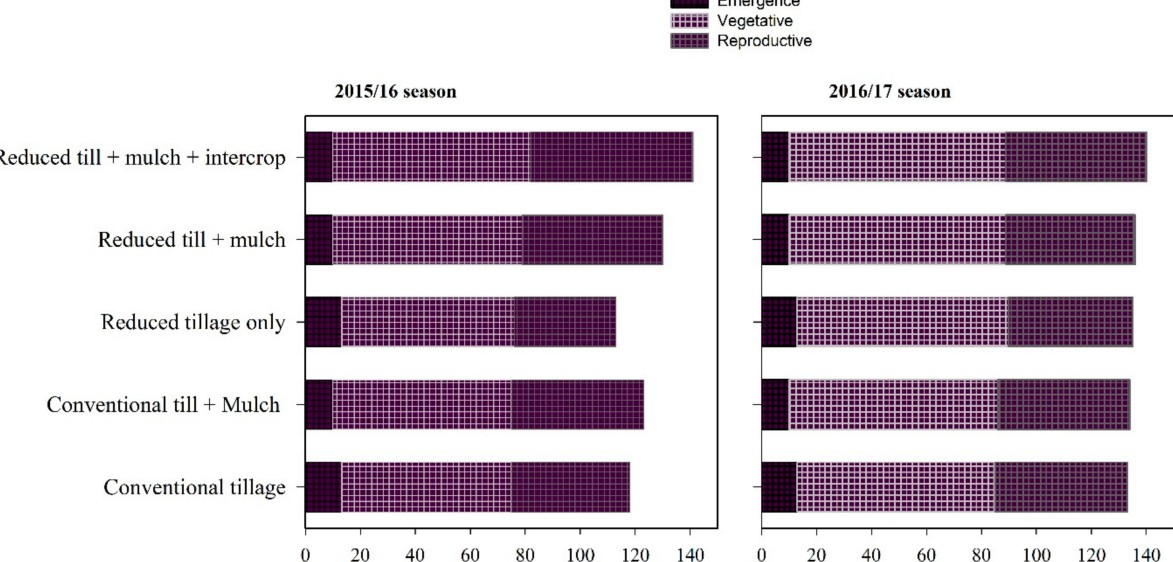

**Figure 3.** Durations of the emergence (from sowing to emergence), vegetative stage (seedling emergence to ear formation) and reproductive stage (silking to physiological maturity) of the finger millet crop with conventional tillage (control), conventional tillage + mulch, reduced tillage only, reduced tillage + mulch and reduced tillage + mulch + intercropping with cowpea treatments in 2015/16 and 2016/17.

**Table 2.** Agronomic performance of finger millet with conventional tillage (control), conventional tillage + mulch (partial CA1), reduced tillage only (partial CA2), reduced tillage + mulch (partial CA3) and reduced tillage + mulch + intercropping with cowpea (full CA) treatments in 2015/16 and 2016/17 in 2015/16 and 2016/17 cropping seasons.

| Treatment | Total Tiller | Productive Tillers | Average Finger No. | Finger Length (cm) | * "Fist-Like" Finger (%) |
|---|---|---|---|---|---|
| 2015/16 | | | | | |
| Control | 2.0 c | 1.3 d | 4.0 d | 4.6 d | 40 d |
| Partial CA1 | 3.1 b | 2.6 c | 4.2 c | 6.6 c | 56 c |
| Partial CA2 | 0.3 d | 0.0 e | 3.3 e | 3.0 e | 24 e |
| Partial CA3 | 7.7 a | 5.3 b | 5.4 b | 7.4 b | 64 b |
| Full CA | 8.3 a | 6.7 a | 6.7 a | 9.1 a | 77 a |
| 2016/17 | | | | | |
| Control | 4.1 | 3.3 b | 5.2 bc | 6.6 c | 55 d |
| Partial CA1 | 3.9 | 3.7 ab | 5.7 b | 7.3 b | 65 c |
| Partial CA2 | 3.7 | 3.3 b | 3.7 c | 5.4 d | 38 e |
| Partial CA3 | 4.3 | 4.1 a | 6.4 a | 9.7 a | 74 b |
| Full CA | 4.7 | 4.4 a | 6.6 a | 10.1 a | 89 a |

Different letters within each column indicate significant differences (*p* < 0.05). * Indicator of well-developed grains according to local farmers.

**Table 3.** Correlation coefficients between yield and total tillers, productive tillers, average number of fingers per panicle, finger length and "fist"-like finger percentage during the 2015/16 2016/17 season.

| Season | Total Number of Tillers | Productive Tiller Number | Average Finger Number | Average Finger Length | "Fist"-Like Finger % |
|---|---|---|---|---|---|
| 2015/16 | 0.73 | 0.83 | 0.91 | 0.89 | 0.88 |
| 2016/17 | 0.80 | 0.70 | 0.68 | 0.65 | 0.86 |

### 3.3. Finger Millet Shoot Biomass and Growth Rate

Starting at 12 weeks after planting, the full CA and partial CA2 treatments resulted in the most and least total biomass, respectively, in both cropping seasons (Figure 4a,b). The shoot biomass for the control, partial CA1 and partial CA3 treatments was not significantly different during either season. At 20 weeks after emergence, the full CA had the highest shoot biomass of 3.8 and 7.0 t ha$^{-1}$ during the 2015/16 and 2016/17 seasons, respectively

(Figure 4a,b). This was over 200% and 60% more than the control treatment during the two respective seasons. The difference in shoot biomass was associated with different growth rates for each treatment (Figure 5a,b). The growth rate under the full CA treatment was the highest during both seasons. Mulching caused a significantly higher growth rate compared with the non-mulched counterpart in both years. At 12 weeks after planting, full CA achieved an 83% higher growth rate than the control during the 2015/16 season. On the other hand, partial CA1, as the second-best option, achieved a 50% higher growth rate than the control. During the 2016/17 season, the corresponding increases were 31% and 16%, respectively.

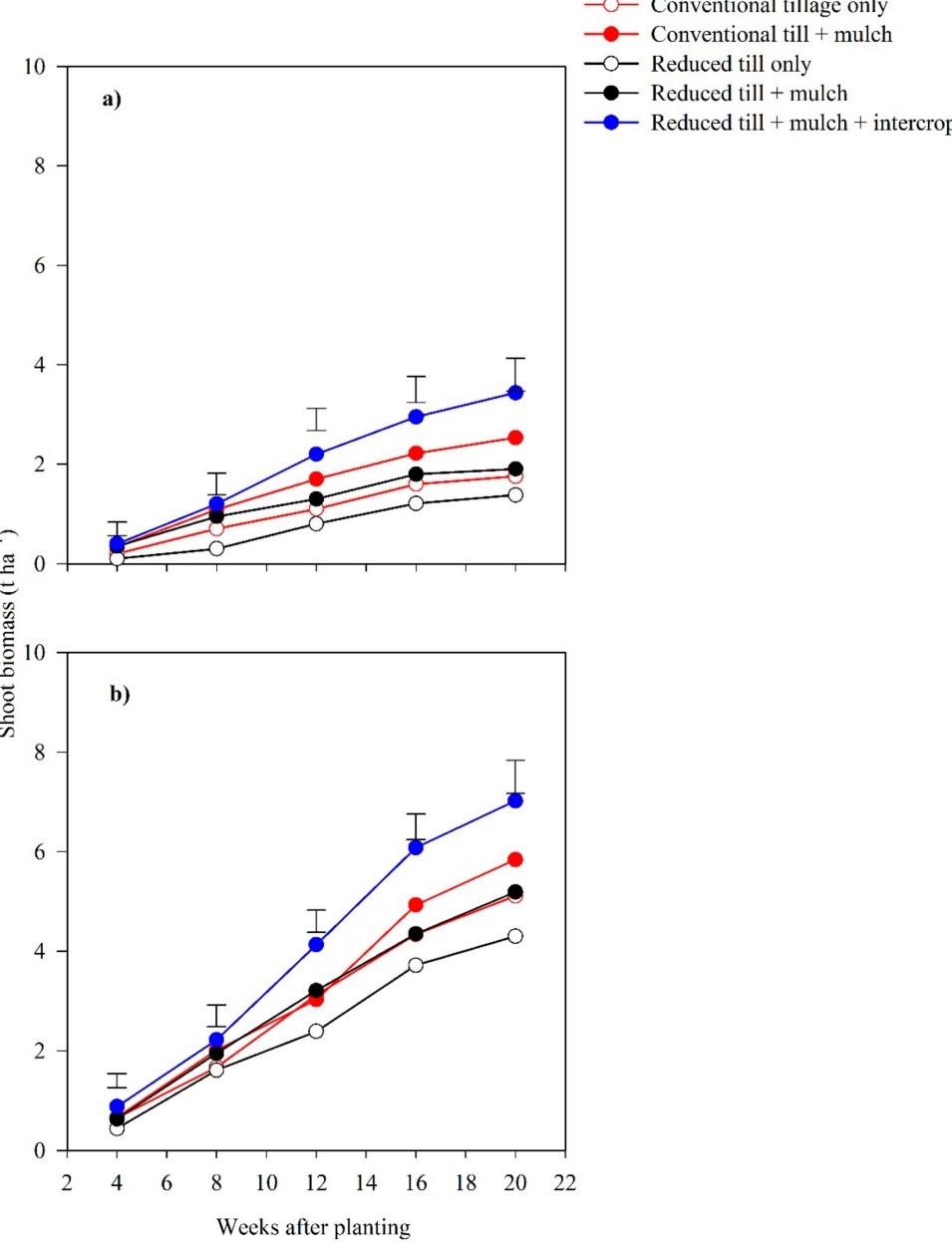

**Figure 4.** Development of finger millet shoot dry biomass with conventional tillage only (control), conventional tillage + mulch (partial CA1), reduced tillage only (partial CA2), reduced tillage + mulch (partial CA3) and reduced tillage + mulch + intercropping with cowpea (full CA) treatments during 2015/16 (**a**) and 2016/17 (**b**) cropping seasons.

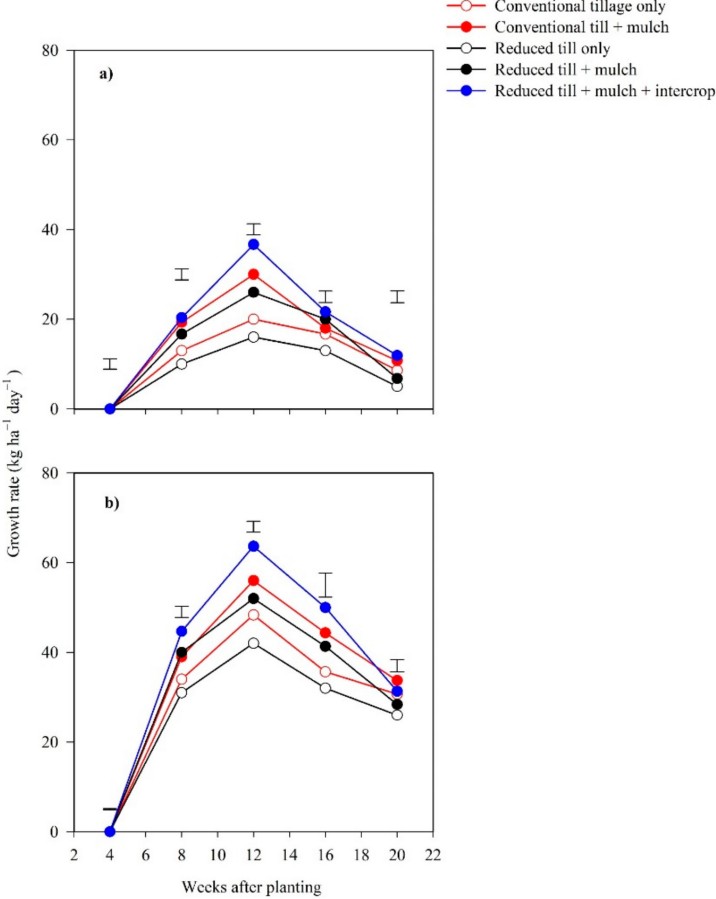

**Figure 5.** Finger millet growth rate with conventional tillage only (control), conventional tillage + mulch (partial CA1), reduced tillage only (partial CA2), reduced tillage + mulch (partial CA3) and reduced tillage + mulch + intercropping with cowpea (full CA) treatments during 2015/16 (**a**) and 2016/17 (**b**) cropping seasons. Error bars represent $LSD_{0.05}$.

*3.4. Finger Millet Grain Yield and Water Use Efficiency*

Overall, season ($p < 0.015$) and treatments ($p = <0.001$) had a significant effect on finger millet grain yield, but their interaction ($p = 0.528$) did not (Table 4). The 2016/17 season, which had fewer incidences of intra-seasonal dry spells, achieved nearly 26% more yield than the 2015/16 season. The full CA treatment had the largest finger millet grain yields of 1.07 and 1.29 t ha$^{-1}$ during the 2015/16 and 2016/17 cropping seasons, respectively (Figure 6). Compared to the control (conventional tillage only), full CA, partial CA3 and partial CA1 increased the total grain yield by 70%, 14% and 17%, respectively, during the 2015/16 cropping season (Figure 7). Similarly, grain yields increases of 53%, 11% and 4%, respectively, were achieved during the 2016/17 cropping season (Figure 7). The partial CA2 achieved 25% less than the control treatment during both seasons.

**Table 4.** Summary of the effect of season (main plot), treatment (sub-plot) and their interaction on finger millet grain yields over two cropping seasons (2015/16 and 2016/17) in on-farm trials in Hwedza, semi-arid Zimbabwe.

| Source of Variation | Degrees of Freedom | *p* Value for Grain Yield |
|---|---|---|
| Season | 1 | 0.015 |
| Treatment | 4 | <0.001 |
| Season × Treatment | 4 | 0.528 |

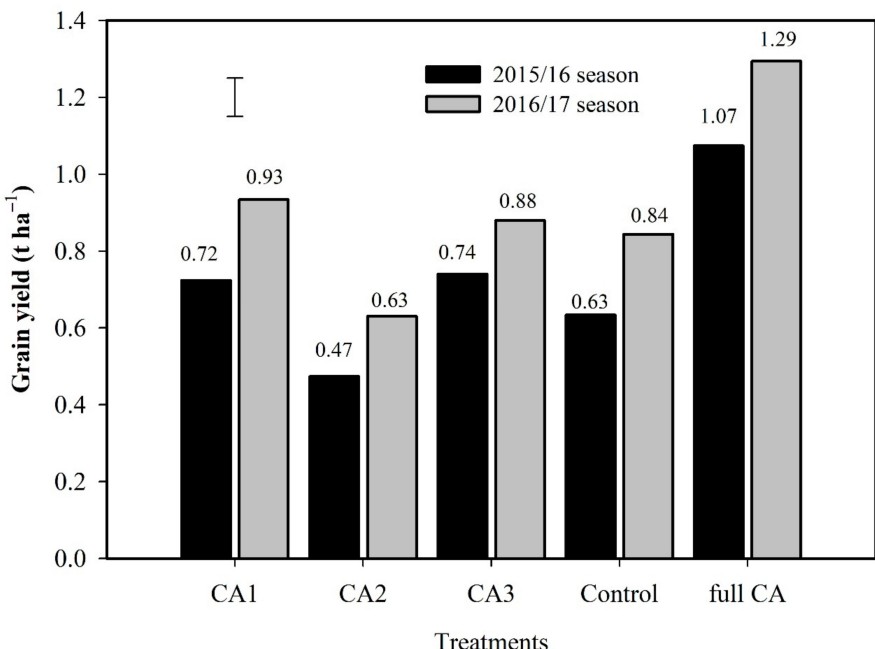

**Figure 6.** Finger millet grain yield with conventional tillage only (control), conventional tillage + mulch (partial CA1), reduced tillage only (partial CA2), reduced tillage + mulch (partial CA3) and reduced tillage + mulch + intercropping with cowpea (full CA) treatments during 2015/16 and 2016/17 cropping seasons. Error bar represents $LSD_{0.05}$.

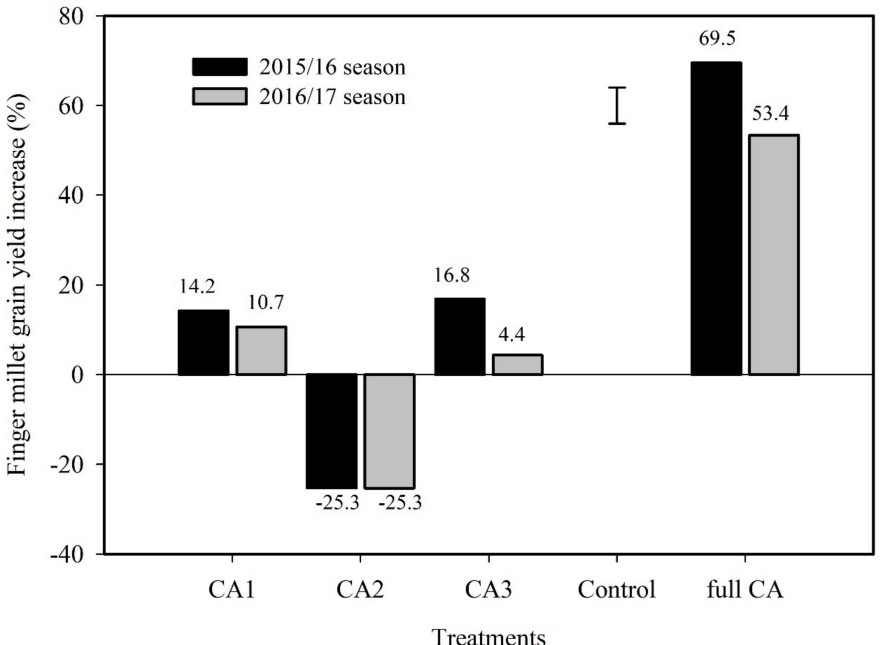

**Figure 7.** Finger grain yield increase with reference to conventional tillage only (control) with conventional tillage + mulch (partial CA1), reduced tillage only (partial CA2), reduced tillage + mulch (partial CA3) and reduced tillage + mulch + intercropping with cowpea (full CA) treatments during 2015/16 and 2016/17 cropping seasons. Error bar represents $LSD_{0.05}$.

Crop water capture, measured as total crop evapotranspiration (ET) and finger millet grain WUE, is presented in Figures 8 and 9, respectively. During both seasons, full CA and CA2 (reduced tillage only treatments) had the highest and lowest ET values, respectively (Figure 8). However, the full CA values were not significantly different to other mulched treatments, especially during the 2015/16 cropping season (Figure 8). Compared to CA2,

crop water capture under full CA was 122% and 63% greater during the 2015/16 and 2016/17 cropping seasons, respectively. Grain WUE values were generally higher during the 2015/16 cropping season, which was characterized by a higher cumulative in-crop rainfall amount and experienced more incidences of intra-seasonal dry spells compared to the 2016/17 season (Figure 9). There was no significant difference among full CA, the control and CA2, which were also the most efficient treatments in the 2015/16 season (Figure 9). Thus, reduced tillage only (CA2) treatment was among the most efficient in terms of finger millet grain WUE despite achieving the lowest ET values.

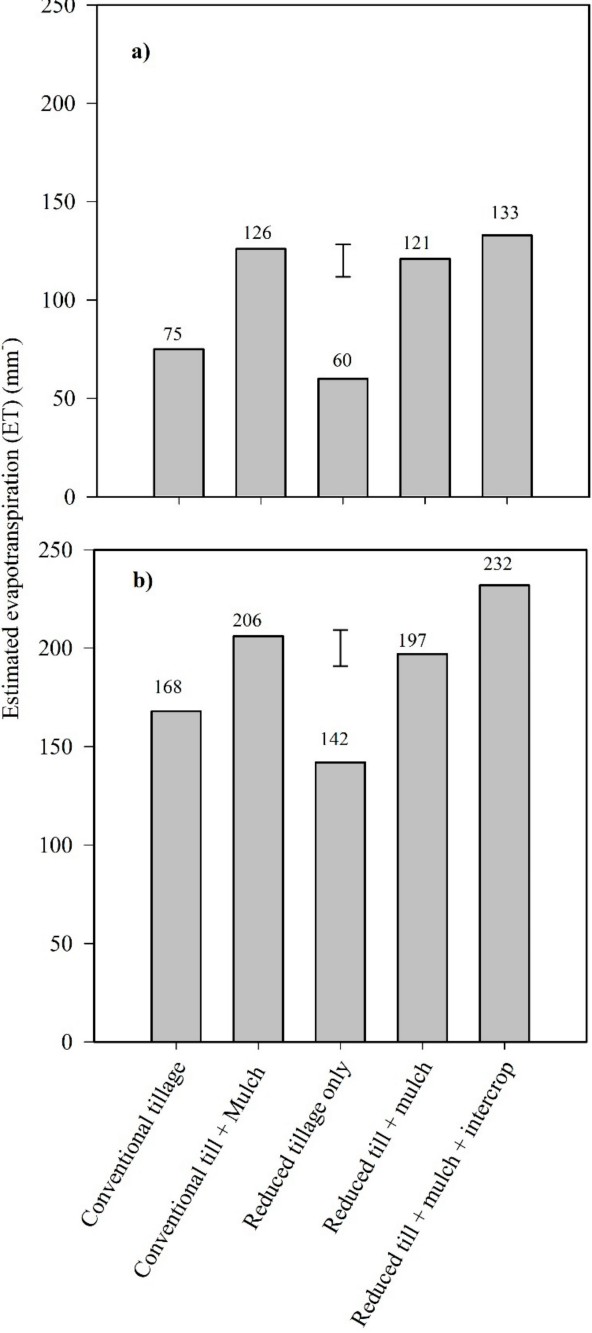

**Figure 8.** Estimated evapotranspiration in conventional tillage only (control), conventional tillage + mulch (partial CA1), reduced tillage only (partial CA2), reduced tillage + mulch (partial CA3) and reduced tillage + mulch + intercropping with cowpea (full CA) treatments under finger millet during 2015/16 (**a**) and 2016/17 (**b**) cropping season. Error bars represent LSD$_{0.05}$.

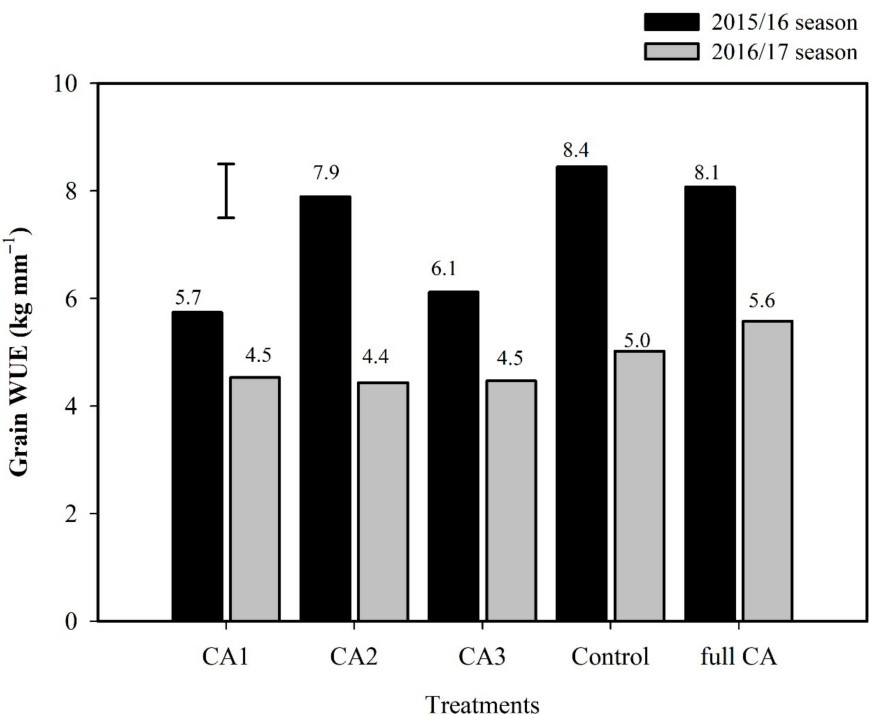

**Figure 9.** Grain water use efficiency in finger millet under conventional tillage only (control), conventional tillage + mulch (partial CA1), reduced tillage only (partial CA2), reduced tillage + mulch (partial CA3) and reduced tillage + mulch + intercropping with cowpea (full CA) treatments under finger millet during 2015/16 and 2016/17 cropping seasons. Error bar represents $LSD_{0.05}$.

## 4. Discussion

### 4.1. Full CA Improves Soil Water Content under Finger Millet

In this study, the full CA practice, in which all the three principles (i.e., reduced tillage, mulching and crop diversity (intercropping)) were applied, greatly improved soil water content under semi-arid conditions. The findings are in agreement with Sidar et al. [27]. Improved soil water content is linked to increased infiltration as well as reduced soil water loss through evaporation and runoff [21,22,27]. Under semi-arid conditions, which are characterized by high evaporative demand in which nearly 70% of the precipitation received is lost through evaporation on bare soil surfaces [21], practices that improve soil moisture conservation are therefore key to any significant improvement in crop yields that can be achieved, especially in rainfed systems. Mulching is the best CA principle which anchors yield benefits in the short to medium term [28,29]. Mulch cover shields the soil surface from solar radiation, thereby retarding soil water loss through evaporation. Intercropping has been observed in this study to enhance soil moisture conservation. Mulching material from plant residues rarely lasts through the growth cycle of the crop due to termites [30]. In particular, the mulching material used during the study period could not last beyond two months after planting. When the mulch cover was present, no significant difference in terms of the amount of soil water content was observed in any of the mulched plots, especially partial CA 1 (conventional tillage + mulch) and full CA (reduced tillage + mulch + intercropping). However, after the two-month period, soil water content in the former experience a significant decrease likely due to mulch cover disappearance due to decomposition and termite attacks, while improved crop ground through intercropping in the latter could explain the high soil water content. Intercropping almost doubles crop ground cover when compared with sole crops, particularly under an additive arrangement [31]. The improved crop ground cover could have provided a "live" mulching role as observed by Mbanyele et al. [31]. Similarly, water conservation of over 60% has been reported in sorghum–cowpea intercropping by Shackel and Hall [32]. While competition between component crops for water resources in intercrops in drought-prone

environments is a common challenge and poses a risk of significant yield reduction or complete crop failure [33,34], a combination with mulch cover from plant residues ("dead" mulch) could have been strategic.

*4.2. Improved Water Availability Enhances Finger Millet Productivity*

Finger millet responded positively to increased water availability, as shown by, firstly, more yield during the 2016/17 cropping season, which received more water and experienced fewer intra-seasonal dry spells. Secondly, treatments that achieved more volumetric water content during both seasons had more yield. In particular, the combination of reduced tillage + mulch + intercrop was the best treatment with regard to the number of productive tillers, longest finger lengths and highest number of fingers, highest biomass accumulated over time and ultimately grain yield. The treatment represents all the three principles of CA. Similar results of such positive responses to improved water management in millets have been reported by other researchers elsewhere who noted that improved water availability improves grain yield through increased numbers of tillers, grain numbers per finger and grain weight [27,35–37]. This is also true for pearl millet, which is perceived to be more drought-tolerant than finger millet [38,39]. The enhanced soil water availability through the combination of "dead" and "live" mulch under reduced tillage reduced water stress throughout the growth cycle of the finger millet, and this agrees with Matsaura et al. [37], who observed that finger millet is sensitive to drought conditions that coincide with both early and late crop growth stages. Therefore, for any substantial yield increase, there is a need to make sure that all the growth stages are proofed against intra-seasonal dry spells.

The benefits of CA in this study were instant. This is contrary to the general observation that CA benefits manifest in the long term (at least 5 years of practice) [40]. The frequent lack of short-term crop yield benefits under CA yield benefits is among the factors suggested to be behind the low adoption by smallholder farmers in SSA [23]. Therefore, full CA practice in finger millet proved a perfect fit. This suggests that the turnaround period of practicing CA, especially with regard to crop yield benefits, may vary with crop type and cropping arrangement. There is also great potential to improve finger millet productivity when CA principles are applied partially compared to conventional practice. Mulch-based conventional tillage was the second-best option to full CA. The treatment performed better than reduced tillage + mulching. It is highly likely that loosened soil under conventionally tilled soil resulted in more infiltration while the presence of mulch cover improves soil water retention through suppressing evaporation. This means that CA may need to be tailored to local conditions.

Improved water availability under full CA practice did not only improve finger millet grain yield but the efficiency with which water was utilized, as shown by highest grain WUE, especially during the 2015/16 cropping season. Soil and water conservation researchers, through experimentation and crop models, have also observed significant improvement in WUE when soil water conservation is improved through practices such as stone bunds [41], half-moons [42], infiltration pits [43], tied ridges and ridge–furrow mulching systems [44]. It was also noted that full CA treatment was not significantly different to the reduced tillage, albeit the later achieved the least in terms of soil water availability and grain yield. There was also higher grain WUE during the 2015/16 crop season that had a smaller cumulative in-crop rainfall amount and experienced more incidences of intra-seasonal dry spells than the 2016/17 cropping season. Such behaviour of improved efficiency under low water resource input in crop plants has been generally observed by Du et al. [45]. This explains why deficit irrigation is recommended for improved WUE in cropping systems [45,46].

The best treatment in this study is within the reach of many smallholder farmers in Zimbabwe, who can improve yields and ultimately improve domestic food supply. The grass used in this study as mulching material is locally available and commonly found around field edges, fallowed degraded fields and along major roads in most smallholder farming areas in Zimbabwe. The grass is highly lignified such that it is less palatable to

livestock. Thus, there are no competing uses as livestock feed as is the case with crop residues that are commonly used in CA. Intercropping, on the other hand, is a common farmer practice. However, the possibility of component crops competing for water resources under intercrop conditions can be a challenge and calls for farmers to be strategic. In this study, mulching was carried out using plant residues as "dead" mulch proved to be one of the strategic options to consider. Soil type has been shown to be a critical factor that influences productivity in CA. There is therefore a possibility that our findings could be different under different soil types, and this factor is worth investigating in the future. Moreover, although full CA in the present study was the most productive option, more could have been done to check if the yield benefits can offset the production costs related to, for example, the labour required to cut and spread the mulching material. Furthermore, including intercropping in conventional mulch-based practices may need to be considered against full CA. Reduced tillage is associated with emergence of perennial weeds, which can be problematic in manage under predominantly manual-labour-based systems in smallholder farming areas of SSA [23].

## 5. Conclusions

We sought to evaluate CA principles when applied individually or in combination with respect to seasonal soil water dynamics and subsequent finger millet productivity in the short term in semi-arid Zimbabwe. Our study findings show the greatest soil water retention was achieved with reduced tillage + mulch + intercrop during the two consecutive seasons. Consequently, the treatment achieved the best finger millet shoot dry biomass and highest grain yield. Thus, effects of CA on finger millet productivity in terms of grain yield and WUE were highest when CA principles were applied in full. The practice therefore presents a potential adaptation strategy for optimizing smallholder finger millet production amidst increasing incidences of intra-seasonal dry spells due to climate change. This study revealed that although finger millet is generally a drought-tolerant crop which tends to perform better than cereals such as maize under semi-arid conditions, the challenging climatic conditions, especially increased incidences of within-season dry spells in recent years, do not present optimum environmental conditions for improved grain yield. Therefore, improved moisture conservation is also critical for the so-called drought-tolerant cereal crops that are mostly grown by farmers in areas characterized by low and erratic rainfall patterns.

**Author Contributions:** V.M.: conceptualization, methodology, formal analysis, investigation, visualization, writing—original draft, writing—review and editing. F.M.: conceptualization, methodology, writing—review and editing, supervision, project administration, funding acquisition. H.N.: conceptualization, methodology, writing—review and editing, visualization, supervision. J.R.: writing—review and editing. P.M.: conceptualization, methodology, writing—review and editing, visualization, supervision, project administration, funding acquisition. All authors have read and agreed to the published version of the manuscript.

**Funding:** This work was funded by the European Union (Contribution Agreement: DCI-Food/2012/304-807) through the Food and Agriculture Organization of the United Nation (FAO) under the "Supporting smallholder farmers in southern Africa to better manage climate related risks to crop production and post-harvest handling (CLIRCS)" project of the Soil Fertility Consortium of Southern Africa (SOFECSA).

**Institutional Review Board Statement:** Not applicable.

**Informed Consent Statement:** Informed consent was obtained from all subjects involved in this study.

**Data Availability Statement:** Not applicable.

**Acknowledgments:** Additional funding for this study was provided by the Government of Zimbabwe through the Ministry of Higher and Tertiary Education Innovation Science and Technology Development and the University of Zimbabwe Vice Chancellor's Award Challenge Fund under the "Future Grains for Africa" program.

**Conflicts of Interest:** The authors declare no conflict of interest.

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
