# Peer review of "Conservation Agriculture in Semi-Arid Zimbabwe: A Promising Practice to Improve Finger Millet (Eleusine coracana Gaertn.) Productivity and Soil Water Availability in the Short Term"

_agriculture, doi:10.3390/agriculture12050622_

Round 1

Reviewer 1 Report

The manuscript entitled "Conservation agriculture in semi-arid Zimbabwe: A promising practice to improve finger millet (Eleusine coracana Gaertn.) productivity and soil water availability in the short term" is an excellent study. The authors have worked well in the preparation of the manuscript. I have some minor comments to improve the structure. 

  • In the intro section add the production and consumption of millet worldwide, in Africa, and zimbabwe.
  • Please justify why the authors used LSD or preferred it over Tukey or Duncan test. 
  • The discussion is short add some mechanistic approaches to justify your points. 
  • Add a subsection as the implementation of this study in real life and its impacts. 
  • Add the study limitations and future research recommendations. 
  • Update references with recent studies. 
  • Minor english language check is needed. 

Author Response

4 March 2022

The Editor

Agriculture Journal MDPI

Re: Re-submission of the revised manuscript: “Conservation agriculture in semi-arid Zimbabwe: A promising practice to improve finger millet (Eleusine coracana Gaertn.) productivity and soil water availability in the short term” – Agriculture 1659879

Dear Editor

On behalf of the co-authors, I would like to thank the editor and reviewers for their valuable comments and contributions, which have helped us to improve our manuscript. The manuscript has been revised according to the comments and suggestions provided by the reviewer. Please find attached list of the changes we have made to the manuscript in response to the reviewer’s comments/recommendations.

Yours Sincerely

Vengai Mbanyele

Soil Fertility Consortium for Southern Africa (SOFECSA)  Research Group, Department of Soil Science and Agricultural Engineering, University of Zimbabwe, P.O. Box MP167, Mount Pleasant, Harare, Zimbabwe

Tel: +263-772670597; E-mail: [email protected] ; [email protected] 

Reviewer 2 Report

This study aims to improve finger millet productivity and soil water availability in Zimbabwe. Overall, the article is relevant, and some issues need to be further clarified. The comments are listed below.

  1. There are many abbreviated technical terms, however, many of them are not defined the first time when they appear in the manuscript which can impede the understanding quite a bit. (e.g., L57 what does SSA means?) Please check the whole manuscript.
  2. L218 by using equation (1), the authors should at least introduce the reason. What if there is groundwater recharge? How about surface runoff? Authors should clarify these concepts, but not only given a simplified equation without any explanation.
  3. L288 c and what? Besides, (a and d) is that right?
  4. Table 2 what does these letters mean?
  5. Table 4 What does the DF mean?
  6. There are only the P values of the results, are there any correlation results?
  7. Figure 8 full CA and CA2 had the highest and least ET values, respectively? And there are no (a) and (b) in the figure.
  8. Figure 9 it seems that CA1, CA2 and CA3 in 2016/2017 season got the same WUE, but compared Figure 6 and Figure 8, CA2 should be the lowest. Is that right? It’s better to write the specific value in each pillar.
  9. There are some grammatical errors in this paper, as well as inconsistent expression, please check the full manuscript very carefully! (e.g., L48, L92, there are some spaces; L125 double “in”; L247 there are no Y in equation (2))

Author Response

4 March 2022

The Editor

Agriculture Journal MDPI

Re: Re-submission of the revised manuscript: “Conservation agriculture in semi-arid Zimbabwe: A promising practice to improve finger millet (Eleusine coracana Gaertn.) productivity and soil water availability in the short term” – Agriculture 1659879

Dear Editor

On behalf of the co-authors, I would like to thank the reviewer for his/her valuable comments and contributions, which have helped us to improve our manuscript. The manuscript has been revised according to the comments and suggestions provided by the reviewer. Below is a list of the changes we have made to the manuscript in response to the reviewer's comments/recommendations.

Yours Sincerely

Vengai Mbanyele

Soil Fertility Consortium for Southern Africa (SOFECSA)  Research Group, Department of Soil Science and Agricultural Engineering, University of Zimbabwe, P.O. Box MP167, Mount Pleasant, Harare, Zimbabwe

Tel: +263-772670597; E-mail: [email protected] ; [email protected] 

Round 2

Reviewer 2 Report

Authors have answered all my questions, I recommend accept.